Graft survival of Pinus engelmannii Carr. in relation to two grafting techniques with dormant and sprouting buds

http://orcid.org/0000-0002-6013-6028 Pérez-Luna Alberto 1 2
http://orcid.org/0000-0002-3284-422X Hernández-Díaz José Ciro 3
http://orcid.org/0000-0002-2341-5458 Wehenkel Christian 3
Simental-Rodríguez Sergio Leonel 1
Hernández-Velasco Javier 1
http://orcid.org/0000-0002-2954-535X Prieto-Ruíz José Ángel 4 jprieto@ujed.mx
1 Programa Institucional de Doctorado en Ciencias Agropecuarias y Forestales, Universidad Juárez del Estado de Durango , Durango, Durango , Mexico
2 Centro de Bachillerato Tecnológico Industrial y de Servicios Número 89, Dirección General de Educación Tecnológica Industrial , Durango, Durango , México
3 Instituto de Silvicultura e Industria de la Madera, Universidad Juárez del Estado de Durango , Durango, Durango , Mexico
4 Facultad de Ciencias Forestales, Universidad Juárez del Estado de Durango , Durango, Durango , Mexico
Eguiarte Luis
Electronic publication date: 2021 Sep 21
Publication date: 2021
Volume: 9
Electronic Location ID: e12182
Received 2021 Feb 9; Accepted 2021 Aug 29
Copyright: © 2021 Pérez-Luna et al.
Copyright year: 2021
Copyright holder: Pérez-Luna et al.
License: This is an open access article distributed under the terms of the Creative Commons Attribution License, which permits unrestricted use, distribution, reproduction and adaptation in any medium and for any purpose provided that it is properly attributed. For attribution, the original author(s), title, publication source (PeerJ) and either DOI or URL of the article must be cited.
License URL: https://creativecommons.org/licenses/by/4.0/

Keywords: Asexual propagation, Apache pine, Top cleft grafting, Side veneer grafting, Grafting survival, Statistical survival analysis models

Funding: This work was funded by the Consejo Nacional de Ciencia y Tecnología (CONACYT) 441054 Consejo de Ciencia y Tecnología del Estado de Durango COCYTED-12/02/18/265 This work was funded by the Consejo Nacional de Ciencia y Tecnología (CONACYT) (441054) and Consejo de Ciencia y Tecnología del Estado de Durango (COCYTED-12/02/18/265). The funders had no role in study design, data collection and analysis, decision to publish, or preparation of the manuscript.

==============================
Developing methods for successfully grafting forest species will be helpful for establishing asexual seed orchards and increasing the success of forest genetic improvement programs in Mexico. In this study we investigated the effects of two grafting techniques (side veneer and top cleft) and two phenological stages of the scion buds (end of latency and beginning of sprouting), in combination with other seven grafting variables, on the sprouting and survival of 120 intraspecific grafts of Pinus engelmannii Carr. The scions used for grafting were taken from a 5.5-year-old commercial forest plantation. The first grafting was performed on January 18 (buds at the end of dormancy) and the second on February 21 (buds at the beginning of sprouting). The data were examined by analysis of variance and a test of means and were fitted to two survival models (the Weibull’s accelerated failure time and the Cox’s proportional hazards model) and the respective hazard ratios were calculated. Survival was higher in the top cleft grafts made with buds at the end of latency, with 80% sprouting and an estimated average survival time of between 164 and 457 days after the end of the 6-month evaluation period. Four variables (grafting technique, phenological stage of the scion buds, scion diameter and rootstock height) significantly affected the risk of graft death in both survival models. Use of top cleft grafts with buds at the end of the latency stage, combined with scion diameters smaller than 11.4 mm and rootstock heights greater than 58.5 cm, was associated with a lower risk of death.

Introduction

An average area of land of 212,000 hectares was deforested annually in Mexico between 2001 and 2018 (Comisión Nacional Forestal (CONAFOR), 2020), mainly due to overexploitation, forest fires, cattle ranching, clandestine logging, adverse weather conditions and a shift in land use from forest to grassland (FAO-CONAFOR, 2009). Between 1970 and 2014 deforestation caused the loss of more than 23,000,000 ha of tropical forest and more than 13,000,000 ha of temperate forest (Secretaría de Medio Ambiente y Recursos Naturales (SEMARNAT), 2018). These figures place Mexico among the first countries worldwide in terms of loss of forest area.

Mexico also has an annual deficit of 11,619,300 m3 of round and manufactured wood, as well as a deficit of 6,535,500 m3 of cellulosic products (Secretaría de Medio Ambiente y Recursos Naturales (SEMARNAT), 2017). The constant demand for timber products from North America indicates that the consumption in Mexico is three times higher than the production (Secretaría de Medio Ambiente y Recursos Naturales (SEMARNAT), 2017). By contrast, the United States of America annually consumes 20% more wood than is produced nationally (Fiedler et al., 2001). Considering forest deterioration, the high demand for wood products and the projected further 33% increase in demand for timber products by 2030 (Perlis, 2009), seeking ways to increase the supply of wood from Mexican forests is imperative.

Commercial forest plantation programs are particularly important in this respect (Martínez & Prieto, 2011). However, there are several deficiencies in the production chain for this option, including lack of high quality genetic germplasm, among other factors (Vargas & López, 2017).

In the state of Durango (Mexico), between 0.9 and 1.3 million hectares of forest cover were disturbed in the period 1986–2012 due to various factors, such as forest fires, land use change, overgrazing and over-exploitation of forests (Novo-Fernández et al., 2018). Fortunately, the state has 1,150,000 ha of land with a high potential for establishing commercial forest plantations (Martínez & Prieto, 2011). Pinus engelmannii Carr. is a promising candidate for this purpose, as it is one of the most economically and environmentally important species, mainly due to the high quality of its wood (Prieto et al., 2004; González-Orozco et al., 2018). Establishment of forest plantations in previously non-forested land can contribute to the conservation of natural forest resources (Ramírez & Simonetti, 2011) and to generating environmental services, such as carbon sequestration (Miehle et al., 2006; Soto-Cervantes et al., 2020), water harvesting, soil retention (Sayer & Elliot, 2005) and landscape improvement (Sabogal, Besacier & McGuire, 2015), among others (Martínez & Prieto, 2011).

The success of commercial forest plantations depends on the use of high-quality genetic material, which can be obtained from asexual seed orchards (Yuan et al., 2016; Stewart et al., 2016; Pérez-Luna et al., 2020a). To establish this type of orchard, superior genotypes must first be multiplied, either by grafting, air layering, cutting or by in vitro propagation (Iglesias, Prieto & Alarcón, 1996; Bonga, 2016). Grafting is the propagation method most commonly used for cloning superior genotypes of forest species (Stewart et al., 2016). Grafts can be made by taking scions from identified superior trees growing in natural environments and grafting them onto rootstocks (branches or rhizomes) from other trees, which are typically grown in a nursery, giving rise to new plants (Jayawickrama, Jett & McKeand, 1991; Muñoz et al., 2013).

Graft success is also influenced by factors such as grafting technique, phenological stage of the scion buds (Jayawickrama, Jett & McKeand, 1991), genetic, anatomical and taxonomic compatibility between scions and rootstocks (Pina & Errea, 2005), efficient nutrition (Mutabaruka, Cook & Buckley, 2015), the grafter’s training and skill, and the graft development environment (Muñoz et al., 2013).

Survival equations, such as the Weibull’s accelerated failure time model and its hazard function, enable estimation of the probability of success and prediction of the survival time, after the last evaluation in the study period (Zhang, 2016). This model has mainly been used in the field of medicine (Chaou et al., 2017) but has recently been applied in studying the success of grafting tree species (Pérez-Luna et al., 2020b).

Another widely used non-parametric survival model, especially in medical science, is the Cox’s proportional hazards model-and the associated hazard ratio-which measures the risk of death of a group of individuals subjected to two or more treatments (Meira-Machado et al., 2013; Gandrud 2015). Pérez-Luna et al. (2019) recently demonstrated the potential use of this model to study the effect of several factors on graft survival and suggested that these modelling techniques are widely applicable in forestry research.

The objectives of the present study were to evaluate the effect of the grafting technique and the phenological stage of the scion buds on the percentage graft survival in P. engelmannii. In addition, the potential effects of another seven variables on graft survival were evaluated. For this purpose, a Weibull’s accelerated failure time model and the associated hazard ratio were tested, also the Cox’s proportional hazards model was fitted to the data. Given the scarce use of these models in the field of forestry, another important objective of this work was the validation of their usefulness for explaining the effect of diverse factors on the survival of P. engelmannii grafts.

Materials & Methods

Rootstock production

The rootstocks were produced in the “General Francisco Villa” forest nursery in Durango, Mexico (23° 58′ 20″ North and 104° 35′ 56″ West, at 1,875 m elevation). During the first year, the rootstocks were grown in polystyrene trays with 77 cavities (each with a capacity of 170 mL); the substrate was a mixture of equal parts of peat moss and composted pine bark. In the second year, each rootstock was transplanted into a 3.5 L black polyethylene bag, which contained equal parts of forest soil and pine bark.

The main physical properties of the substrate were as follows: electrical conductivity, 0.16 mmhos cm−1; pH 6.04; organic matter content, 3.8%; and organic carbon content, 2.2%. The total porosity, aeration porosity and the water-holding capacity of the substrate were calculated with the methodology proposed by Landis et al. (1990), yielding values of 82%, 44% and 38% respectively. Analysis of the chemical properties of the substrate revealed values of 1.02% for nitrogen-ammonium (N-NH4), 16.73% for phosphorus (P) and 1.8% for potassium (K).

At age four years (July 2018), and six months before grafting, the rootstocks were transplanted to 5 L containers (bags), to favour reactivation of the root system. The substrate was forest soil, and to each container we added 50 g of MulticoteTM 6: 18-6-12 (N-P-K) + micronutrients, which is a slow-release fertilizer manufactured by the Haifa GroupTM, with a longevity of six months at an average temperature of 21 °C. The rootstocks were watered every three days, with two L of water per plant.

Collection of scions

The scions were taken from trees in a commercial plantation of P. engelmannii, in the Ejido Aquiles Serdán, Durango, Mexico (location 23° 53′ 39″ North and 104° 33′ 44″ West and 1,898 m elevation). The plantation was five years old when the scions were collected and showed good adaptation and growth. The selected donor trees were of average height three m and diameter 7 cm at the base of the stem. The scions were collected the day before grafting and placed in layers in 72 L plastic boxes; each layer of scions was covered with sawdust wetted with a solution of three g L−1 of Captán® fungicide, to prevent fungal damage.

Grafting

Grafting was carried out in the nursery of the Institute of Forestry and Wood Industry of the Universidad Juárez del Estado de Durango (ISIMA-UJED), in a greenhouse of dimension 6 × 8 × 3 m (width × length × height), covered with white plastic (caliber 720). To prevent excessive temperatures being reached and to promote adequate relative humidity, two shading meshes (providing 50% and 70% cover) were placed over the greenhouse, 30 cm above the plastic cover. In addition, two air conditioning systems were installed at each end of the greenhouse. The average temperature was maintained at 22 °C (maximum, 26 °C and minimum, 7 °C) and the relative humidity fluctuated between 72% and 82%.

Treatments evaluated

The grafting methods evaluated were the side veneer technique described by Muñoz et al. (2013) and Pérez-Luna et al. (2019) and the top cleft technique described by Muñoz et al. (2013). For each technique, two phenological stages of the buds were tested; half of the grafts were made with scion buds at the end of latency (grafted on January 18, 2019) and the other half were made with the scion buds at the beginning of the sprouting stage (grafted on February 21, 2019). In total, four treatments were evaluated, with 120 grafts (60 of side veneer and 60 of top cleft). Thirty grafts of each phenological condition of the buds were made with each grafting technique.

For the side veneer graft, a longitudinal cut of six cm was made on one side of the scion and a wedge of approximately one centimetre was formed with a cut at the lower end of the other side (Fig. 1A). A lateral cut was made in the rootstock, of the same length as the cut on the scion, and a one-cm slit was left at the end of the cut on the rootstock, into which the scion wedge was inserted and tied (Fig. 1B).

Figure 1 Grafting techniques and graft protection.

(A) Preparation of the scion used for a side veneer graft; (B) a tied side veneer graft; (C) preparation of rootstock for a top cleft graft; (D) a tied top cleft graft; (E) a plastic bag being placed over the grafted area containing water to generate the wet microclimate; (F) protection of the grafted area with a kraft paper bag, to protect the area from direct solar radiation. Photo credit: Alberto Pérez-Luna.

For the top cleft graft, two longitudinal 6 cm cuts were made at the bottom and opposite sides of the scion, forming a wedge shape; the central leader was eliminated from the rootstock, and a central cut (fissure) was made to an approximate depth of 6 cm, into which the scion wedge was inserted (Fig. 1C). The two components of the graft were then tied together (Fig. 1D).

The grafts were tied (both techniques) with transparent rubber tape and sealed with vinyl paint mixed with 3 g L−1 of Captán® fungicide, to prevent pathogens entering the graft union. Finally, a 5 L transparent plastic bag was placed around the grafted area, into which 1 L of water was poured to generate a humid microenvironment (Fig. 1E); in addition, each graft was covered with a kraft paper bag, which provided protection from solar radiation (Fig. 1F).

The grafts were watered every three days with plain water. In addition, from the 3rd month of evaluation, Promyl® fungicide was added to the irrigation water (two g L−1) every eight days, to prevent fungal damage. To compensate for possible nutrient deficiency due to the loss of mycorrhiza caused by fungicide application, fertigation was applied during the six-month evaluation period by adding Ultrasol® Triple 19 (N-P-K) + MgO + micronutrients, a water-soluble fertilizer manufactured by Soquimich commercial, at a dose of 3 g L−1 to the water. The solution was applied in a 7 L manual watering can.

Experimental design and variables evaluated

The treatments were applied in a 2 × 2 factorial experimental design (two grafting techniques × two phenological stages of the scion buds). Each experimental unit consisted of 10 grafts, with three repetitions per treatment. Before grafting, seven internal grafting variables of the scion and rootstock were measured (Table 1).

Table 1 Values of graft variables measured in scions, buds and rootstocks of Pinus engelmannii Carr.

Graft variable	Minimum	Mean ± standard deviation	Maximum	Coefficient of variation	
Length of scion (cm)	8.5	15.1 ± 3.7	32.0	0.24	
Diameter of scion (mm)	7.0	11.4 ± 2.6	18.8	0.22	
Length of bud (cm)	0.5	2.8 ± 1.1	5.5	0.39	
Height of rootstock (cm)	61.0	88.2 ± 14.2	129.0	0.16	
Diameter of rootstock at root crown (mm)	13.3	33.4 ± 5.8	46.7	0.17	
Height of graft (cm)	9.0	58.5 ± 26.8	112.0	0.45	
Diameter of rootstock at graft height (mm)	7.0	18.1 ± 6.1	32.7	0.33	

Sprouting and survival of the grafts were evaluated monthly for six months, and the data were examined by the Kolmogorov–Smirnov test to evaluate the normality of the variables evaluated. Analysis of variance (ANOVA) was used to detect potential significant differences between treatments. When significant differences were indicated and the variables were normally distributed, a post hoc Tukey’s means test was carried out, with an initial confidence interval of 95% (α = 0.05). The Bonferroni correction was applied to reduce the probability of making a type I error (Garamszegi, 2006; Napierala, 2012); the corrected significance value was α* = 0.0125. In addition, Student’s t-tests were used to determine any significant differences in graft survival due to different levels of the independent factors, i.e., side veneer grafts vs top cleft grafts and scions with buds at the end of latency vs scions with buds at the beginning of sprouting. In order to reduce the effect of extreme observations, before performing the analysis of variance and Student’s t tests (Burbidge, Magee & Robb, 1988) the survival value of each treatment was transformed, by calculating the square root of the sine function of the survival quotients.

Finally, the Weibull’s accelerated failure time model was fitted to the data to predict the estimated graft survival time after the evaluation period, and the associated hazard ratio was also calculated. Other hazard ratios were estimated using the Cox’s proportional hazards model. All statistical analyses were implemented in the “Survival” package in the free R software (R Development Core Team, 2018).

Fitting the Weibull’s accelerated failure time model and the hazard ratio

To fit the accelerated failure time model to the graft survival data, the following independent variables were used: grafting technique, phenological stage of the scion buds and the seven grafting variables (Table 1). To detect the significant variables affecting the estimated survival time of the grafts, stepwise regression was applied using the “StepReg” package in R (R Development Core Team, 2018). The Weibull´s accelerated failure time model is defined as follows:

(1) Ln(T)=α+δxi+σε

where Ln (T) is the natural logarithm of the mean survival time (T) after the study, i.e., at time (T) at least one death may occur among the grafts that were alive at the end of the evaluation period; α is a scale parameter of the model, δ is the coefficient of the explanatory variable, σ is the shape parameter of the model and ε is the error of the distribution function (George, Seals & Aban, 2014; Zhang, 2016). The Weibull’s hazard ratio is described as follows:

(2) HRW=(e−β)λ−1

where HRW is the Weibull’s hazard ratio, which represents the increase or decrease in the risk of death of grafts as a function of the independent variables; β is a coefficient that indicates the effect of a given independent variable (from the same xi variables used in Eq. (1)). The value of the dependent variable is not used in Eq. (2), because the Weibull hazard ratio is considered to be “constant for each variable”, i.e., there is only one hazard ratio value for each explanatory variable in the model (Igl, 2018). The β coefficient is interpreted as follows: if β is positive the risk of death decreases as the value of the independent variable decreases, and if β is negative, the risk of death decreases as the value of the independent variable increases (Carroll, 2003). Finally, λ is a shape parameter of the model. If λ > 1 the hazard ratio increases, and if λ < 1 the hazard ratio decreases (Zhang, 2016). The β and λ parameters were calculated with flexible parametric regression, which is useful for modelling survival from a time of origin to the instant at which an event occurs (life or death) (Igl, 2018; Pérez-Luna et al., 2020b).

The Weibull hazard ratio can take values ranging between 0 and ∞ (Ruíz, 2012). A hazard ratio of 0 indicates that the risk of death decreases by 100% [(0 − 1) × 100]. A hazard ratio of > 0 but < 1, for example HRW = 0.5, indicates that the risk of death due to the effect of a variable is reduced by 50% [(0.5 − 1) × 100] when the level of that particular variable changes. On the other hand, if the hazard ratio is 1.0, the risk of death does not vary due to the effect of changes in some particular variable [(1 − 1) * 100]; if the hazard ratio takes a value of two the risk of death doubles, i.e., the risk of death increases by 100% due to the effect of changes in the explanatory variable [(2 − 1) × 100] (Hilsenbeck et al., 1998; Spruance et al., 2004). Interpretation of the hazard ratio is similar for values greater than two (Ruíz, 2012). It is important to take into account that the Weibull hazard ratio is calculated individually for each dependent variable in the model (Zhang, 2016).

For fitting these models, dummy variables were used to code the dependent variable (survival) and the independent variables (grafting technique and phenological stage). Therefore, a live graft was coded as zero and a dead graft as one (censor variables). The coding for the grafting technique was one for side veneer grafts and two for top cleft grafts; the phenological stage was coded as one for cuttings with buds at the end of dormancy and two for cuttings with buds at the beginning of sprouting. For more details on the Weibull’s accelerated failure time model and its hazard ratio, see Pérez-Luna et al. (2020b).

Fitting the Cox’s proportional hazards model

The variables shown in Table 1 were used to fit this model, and the most significant variables were selected by stepwise regression. The Cox’s proportional hazards model used for calculating hazard ratios is as follows:

(3) HRC=e(φ1xi1+…+φkxik)

where HRC is the Cox’s hazard ratio expressed as a function of the independent variables (xi) with which the model is fitted in order to predict the increase or decrease in the risk of death of the individuals under study (grafts). φ represents the fit parameters of the model, up to the k-th independent variable. If a given φ is positive, the hazard ratio increases when the value of the corresponding x increases (decreasing the probability of survival); if the value of a φ is negative, the hazard ratio decreases when the respective x increases (increasing the probability of survival). The range of values and the interpretation of the Cox’s hazard ratio are the same as for the Weibull’s hazard ratio. The Cox’s hazard ratio value is calculated globally, i.e., by including all the independent variables at one time in the HRC equation (Ata & Sözer, 2007).

Several authors recommend the use of hazard ratios derived from the Weibull’s model, as long as the shape parameter of the model (λ), which must be calculated before using the hazard ratio (Lee & Wang, 2003; Lawless, 2011), is known; however, its counterpart, the hazard ratio of the Cox’s proportional hazards model, does not depend on the evaluation time or predicted survival and therefore has the advantage of being less restrictive than the HRW (Cox, 1972).

To calculate the hazard ratio for graft survival using the Cox’s proportional hazards model, dummy variables, including censor variables (dead grafts), were also used, so that the coding for the dependent variables was the same as in the Weibull model. Further details on the use of this model to assess graft survival are given by Pérez-Luna et al. (2019).

Results

The results of the survival analyses were congruent with the estimated hazard ratios, as can be seen in the results described below.

Survival

A total of six months after grafting, the average survival of the grafts made with the top cleft and side veneer techniques were 56.7% and 18.3%, respectively, with significant differences between them. There were also differences in regard to the phenological stages of the buds, analyzed as individual factors (Table 2 and Fig. 2), yielding 50% survival in grafts with buds at the end of latency and 25% survival in the grafts with buds at the beginning of sprouting.

Table 2 Results of the analysis of variance of graft survival.

Factor	Degrees of freedom	Square mean	F value	p < F	
Grafting technique	1	0.44	23.00	0.001*	
Phenological stage of the buds	1	0.19	9.78	0.014	
Interaction of grafting technique x phenological stage of the buds	1	0.14	7.35	0.026	
Note:

* Significance after Bonferroni correction (α = 0.0125).

Figure 2 Graft survival at the end of the six-month evaluation period.

(A) Grafting technique; (B) phenological stage of the buds. Statistical results based on Student’s t tests. Different letters indicate significant differences between treatments (α = 0.05), before Bonferroni correction. The whiskers represent the standard error of each treatment.

The analysis of variance indicated significant differences (before the Bonferroni correction) due to the effect of these two treatments (Table 2); however, after Bonferroni correction only the grafting technique was statistically significant. Furthermore, the Tukey test revealed that the best interaction was the combination of top cleft grafts with the scion buds at the end of the latency, with 80% survival, while the treatment yielding lowest survival (16.7%) was the side veneer grafts with the scion buds at the beginning of sprouting (Fig. 3).

Figure 3 Survival of Pinus engelmannii grafts by treatment, at age six months.

Different letters indicate significant differences between treatments (α = 0.05), before Bonferroni correction. The whiskers represent the standard error of each treatment.

Fitting the Weibull’s accelerated failure time model and the hazard ratio

The stepwise regression indicated that survival can be described with only four independent variables, using the Weibull’s accelerated failure time model for which the lowest value of the Akaike information criterion (AIC) was achieved (Table 3). The model fit was highly significant, even after Bonferroni correction (p < 0.0001), for predicting the mean graft survival time after finishing the six-month evaluation period.

Table 3 Selection of the most significant variables by stepwise regression, for fitting the Weibull accelerated failure time model.

Variables	Number of variables	AIC	
GT, PSB, HR, DRRC, LS, DS, HG, DRGH, LB	9	910.3	
GT, PSB, HR, DRRC, LS, DS, HG, DRGH	8	908.3	
GT, PSB, HR, DRRC, LS, DS, HG	7	907.0	
GT, PSB, HR, DRRC, LS, DS,	6	906.2	
GT, PSB, HR, DRRC, DS	5	905.4	
GT, PSB, HR, DS	4	904.8	
Note:

AIC, Akaike information criterion; GT, Grafting technique; PSB, Phenological stage of the buds; HR, Height of rootstock; DRRC, Diameter of rootstock at root crown; LS, Length of scion; DS, Diameter of scion; HG, height of graft; DRGH, diameter of rootstock at graft height; LB, length of bud.

In addition, all parameters estimated for the fitting of this four-variable model were significant for the original critical level proposed at the beginning (p < 0.05), and only the phenological stage of the buds was not significant after the Bonferroni correction (p < 0.0125) (Table 4).

Table 4 Estimated parameters for fitting the Weibull accelerated failure time model, to estimate the survival time of the grafts after the evaluation period.

Parameter	Estimator	|z|	p < |z|	
Intercept (α)	4.63375	11.11	< 0.0001*	
Coefficient for “grafting technique” (δ1)	0.66768	5.32	< 0.0001*	
Coefficient for “phenological stage of buds” (δ2)	−0.23356	−2.03	0.0425	
Coefficient for “diameter of scion” (δ3)	−0.10703	−4.36	< 0.0001*	
Coefficient for “height of rootstock” (δ4)	0.01440	3.11	0.0018*	
Model scale (σ)	0.46900	−7.87	< 0.0001*	
Note:

* Significance after Bonferroni correction (α = 0.0125).

Application of the accelerated failure time model (Eq. (1)) to the data showed that the estimated time of survival, after finishing the evaluation period, was greater for those grafts produced by the top cleft technique (x = 2) and with scions’ buds at the end of latency (x = 1). In addition, the estimated survival time was greater when the diameter of the scion was less than 11.4 mm (x ≤ 1) and the height of the rootstock greater than 58.5 cm (x ≥ 1). The longest estimated average survival time, after the observation period (6 months), was 457 days, while the shortest estimated average survival time was 164 days.

The estimates depend on the combinations of the values of each independent variable, and these results represent the estimated time during which at least one death could occur among the grafts that were alive at the end of the evaluation period (Zhang, 2016; Pérez-Luna et al., 2020b).

The estimated parameters (calculated by regression) for fitting the Weibull hazard function (Zhang, 2016) are shown in Table 5. A negative sign of the β coefficient indicates that when the variable “grafting technique” increases by one unit, from x = 1 (side veneer graft) to x = 2 (top cleft graft), the risk of death of at least one graft will be reduced by 76% [(HRW − 1) × 100 = (0.24 − 1) × 100 = −76%]; that is, the risk of death decreases by 76% when grafting with the top cleft technique. Interpretation of the negative value of the β estimator of the variable “rootstock height” is the same as the case described above; therefore, for rootstocks of height greater than 58.5 cm, the risk of death of the grafts is reduced by 4% [(HRW − 1) × 100 = (0.96 − 1) × 100 = −4%]. On the other hand, the positive signs of β for the variables “phenological stage of buds” and “diameter of scion” indicate that by increasing the respective variable by one unit, the risk of death of the grafts will increase according to the value obtained for each hazard ratio; thus, in the case of the “phenological stage of buds” for x = 2 (grafts with buds at the beginning of sprouting), the risk of death increased by 64% [(HRW − 1) × 100 = (1.64 − 1) × 100 = 64%] when the phenological stage is x = 1 (grafts with buds of scions at the end of dormancy). Finally, for scions of diameter greater than 13.38 mm, the risk of death of the grafts increased by 25% [(HRW − 1) × 100 = (1.25 − 1) × 100 = 25%].

Table 5 Estimated parameters and results of the calculation of the Weibull hazard function, in relation to the risk of death of the grafts.

Parameter	Estimator	HRW value	p < |z|	
Shape (λ)	5.11 × 10−05	na	< 0.0001*	
Coefficient of “grafting technique” (β1)	−1.42	0.24	< 0.0001*	
Coefficient of “phenological stage of buds” (β2)	4.98 × 10−01	1.64	0.0425	
Coefficient of “diameter of scion” (β3)	2.28 × 10−02	1.25	< 0.0001*	
Coefficient of “height of rootstock” (β4)	−3.06 × 10−03	0.96	0.0018*	
Notes:

HRw, Weibull hazard ratio; na, there is no hazard ratio value for the shape parameter;

* Significance after Bonferroni correction (α = 0.0125).

Fitting the Cox’s proportional hazards model

Using stepwise regression, the variables that best explained the hazard ratio in graft survival were also selected by fitting the Cox proportional hazards model and it was confirmed that the grafting technique, the phenological stage of the buds, the rootstock height and the scion diameter were the outstanding variables for the best fit (Table 6).

Table 6 Selection of variables by stepwise regression to adjust the Cox proportional hazards model.

Variables	Number of variables	AIC	
GT, PSB, HR, DRRC, LS, DS, HG, DRGH, LB	9	615.4	
GT, PSB, HR, DRRC, LS, DS, HG, DRGH	8	613.4	
GT, PSB, HR, DRRC, LS, DS, HG	7	612.1	
GT, PSB, HR, DRRC, LS, DS,	6	611.2	
GT, PSB, HR, DRRC, DS	5	610.3	
GT, PSB, HR, DS	4	609.6	
Note:

AIC, Akaike information criterion; GT, Grafting technique; PSB, Phenological stage of the buds; HR, Height of rootstock; DRRC, Diameter of rootstock at root crown; LS, Length of scion; DS, Diameter of scion; HG, height of graft; DRGH, diameter of rootstock at graft height; LB, length of bud.

The variables that best explained the hazard ratio of the Cox’s proportional hazards model were the same variables selected for fitting the Weibull’s accelerated failure time model, and although Tables 3 and 6 are similar, note that the value of the Akaike’s information criterion was lower in Table 6 (AIC = 609.6) than that shown in Table 3 (AIC = 904.8). Therefore, the Cox’s proportional hazards model performed better than Weibull’s model for predicting the survival probability. The estimators calculated for fitting the Cox’s model are shown in Table 7.

Table 7 Parameters estimated in fitting the Cox proportional hazards model to calculate hazard ratios in relation to graft survival.

Parameter	Estimator	|z|	p < |z|	
Coefficient of “grafting technique” (φ1)	−1.37692	−5.32	< 0.0001*	
Coefficient of “phenological stage of buds” (φ2)	0.54683	2.24	0.0024*	
Coefficient of “ diameter of scion” (φ3)	0.21635	4.12	< 0.0001*	
Coefficient of “height of rootstock” (φ4)	−0.02964	−3.03	0.0024*	
Note:

* Significance after Bonferroni correction (α = 0.0125).

The hazard ratio values (Table 8) were calculated using Eq. (3). It was estimated that the lowest value of the hazard ratio of grafts death (0.18) would be obtained for the top cleft grafts with the scion buds at the end of the dormancy period, using rootstocks taller than 58.5 cm and scions with diameter smaller than 11.4 mm; this value implies that the risk of death would decrease by 82% [(HRC − 1) × 100 = (0.18 − 1) × 100 = −82%] when grafts are produced with this combination of variables, relative to the hazard ratio obtained for other combinations. On the other hand, the highest estimated hazard ratio corresponded to the side veneer grafts using scions with buds at the beginning of sprouting, for rootstocks shorter than 58.5 cm and scions with diameter greater than 11.4 mm, taking a value of 2.0, which indicates that the risk of death under this combination increases by 100% [(HRC − 1) × 100; (2 − 1) × 100 = 100%], relative to the other combinations.

Table 8 Estimated values for the Cox hazard ratio in relation to graft survival.

Combination of variables	Cox hazard ratio	
Graft technique	Phenological stage of buds	Rootstock height greater than 55.8 cm	Rootstock height less than 55.8 cm	Scion diameter greater than 11.4 mm	Scion diameter less than 11.4 mm	
Side veneer	End of latency		X		X	0.75	
X			X	0.71	
	X	X		1.15	
X		X		1.09	
Start of sprouting		X		X	1.29	
X			X	1.22	
	X	X		2.00	
X		X		1.88	
Top cleft	End of latency		X		X	0.19	
X			X	0.18	
	X	X		0.29	
X		X		0.27	
Start of sprouting		X		X	0.33	
X			X	0.31	
	X	X		0.50	
X		X		0.47	
Note:

The maximum and minimum values of the Cox hazard ratio are indicated in bold.

Discussion

In this study, a higher survival of top cleft grafts was observed (56.7%) than in side veneer grafts (18.3%); the two types of grafts were maintained under climate-controlled conditions within the greenhouse. Temperature control during coniferous grafting is important to increase the survival probability, as observed by Blada & Panea (2011), who found that callus formation in side veneer grafts of Picea pungens Engelm var. glauca Regel was favoured by temperatures between 20 and 25 °C.

In a recent study in Durango, Mexico, Pérez-Luna et al. (2019), performed side veneer grafting with 5 to 7-year-old rootstocks of Pinus engelmannii, under greenhouse conditions without automatic climate control, and reported survival of only 22.5% 6 months after grafting, with no significant differences between grafting using scions with buds at the end of dormancy or at the beginning of sprouting. These low survival results were attributed by the authors to the high temperatures registered in the greenhouse during the months of March to May (maximum recorded, 42.6 °C), and to the low relative humidity in the same months (on average, 38%). However, in the present study, the survival obtained in the side veneer grafts, were similar to obtained for Pérez-Luna et al. (2019), indicating that climate control appears to be ineffective in this regard.

On the other hand, the higher survival of the top cleft grafts than of the side veneer grafts achieved in the present study may be related to the more favourable (controlled) environment (maximum temperature of 26 °C and minimum relative humidity of 72%) inside the greenhouse.

However, low temperatures also usually strong affect the survival of coniferous grafts, as observed by Cuevas Cruz (2014), who reported 100% mortality in top cleft and side veneer grafts of Pinus leiophylla Schiede ex Schltdl. et Cham for a minimum temperature of −5.8 °C.

In side veneer grafts of Pinus greggii Engelm var. australis Donahue & López in Veracruz, survival was greater than 60% after 3 months, which was partially attributed to the fact that both the seedlings with which the rootstocks were produced and the grafted scions were obtained from the same geographic location (Alba-Landa et al., 2017). In the present study, the scions and rootstocks were of different origin, which may contribute to explaining the low percentage of successful grafts with this technique. In this respect, some studies did not observe any effect of the origin on the success of grafting; e.g., Aparicio et al. (2009) reported that the provenance of the scions did not influence the survival of terminal fissure grafts of Austrocedrus chilensis (D. Don) Pic Serm & Bizzarri; however, Villaseñor & Carrera (1980) reported that the provenance of the scions had a significant effect on the success of the top cleft and side veneer grafting of Pinus patula Schl. et Cham.

The genetic compatibility between the scion and the rootstock is also a determining factor in grafting success (Lott et al., 2003; Hibbert-Frey et al., 2011); in this regard, it is important to consider the results of recent studies (Wehenkel et al., 2020; Simental-Rodríguez et al., 2021), which indicate large genetic differences and even hybridizations between close populations of the genus Pinus in northern Mexico.

On the other hand, Wendling, Stuepp & Zuffellato-Ribas (2016) found that grafting of Araucaria angustifolia (Bertol.) Kuntze was effective with the chip budding technique (which is similar to the side veneer grafting regarding the cut made in the rootstock), reporting almost 40% survival at 130 days after grafting.

Mencuccini et al. (2007) recommend using rootstocks younger than two years for grafting Pinus species. Dorman (1976) pointed out that the sprouting of side veneer grafts in Pinus species was significantly lower when rootstocks older than 3 years were used. In the present study, the rootstocks were older than 4 years and the survival of side veneer grafts was low (18.3%) despite the fact that the rootstocks used in both grafting techniques were of the same age; however, the survival of top cleft grafts (56.7%) can be considered acceptable. In another study, Zhang & Tang (2005) reported 50% survival in top cleft grafting of Pinus ponderosa Douglas ex C. Lawson with two-year-old rootstocks. In the top cleft graft of Pinus elliottii Engelm., a survival rate of 30% was reported for grafting performed with 2-to 3-year-old rootstocks (Mergen, 1955).

Other authors have also achieved good survival when grafting conifers with the top cleft technique. For example, Almqvist (2013a); Almqvist (2013b) reported survival rates of 75.0% and 84.7% in two Pinus sylvestris L. top cleft grafting experiments. In Abies fraseri (Pursh) Poir grafts, Hibbert-Frey et al. (2011) reported 86% survival with the same technique. Similarly, Singh (1992) reported good success when grafting Pinus gerardiana Wall, using the top cleft technique, with 70% survival.

Villaseñor & Carrera (1980) reported survival of 63.0% in top cleft grafts of Pinus patula in Mexico, using scions with dormant buds. Likewise, Świerczyński et al. (2020) achieved survival greater than 80% for side veneer grafts on Pinus mugo Turra, established in winter (using scions with dormant buds). Survival greater than 60.0% was reported for top cleft grafts of Araucaria angustifolia, in Brazil, using scions with dormant buds (Gaspar et al., 2017). On the other hand, when grafting A. angustifolia, Zanette, Oliveira & Biasi (2011) reported only 20% and 0% survival in grafts with buds at the beginning of sprouting (spring) and during full sprouting (summer), respectively. These results are consistent with those obtained in the present study and, although different species of the order Pinales were used, they must share certain characteristics in terms of their phenological functioning (Bodnar et al., 2015).

To analyze graft survival, Pérez-Luna et al. (2020b) fitted the Weibull’s accelerated failure time model to data on side veneer grafts of Pinus engelmannii, estimating an average survival time of 154 days after the end of the 6-month evaluation period. The higher values of potential survival time (up to 457 days) observed in the present study can be attributed to the automated temperature and environmental humidity controls in the greenhouse where the grafting was conducted in the present study. By contrast, only two shading meshes (providing 70% and 50% light retention) were placed above the greenhouse in the previous study, to reduce the temperature inside.

The negative values of the estimators of the coefficients of the explanatory variables in Table 7 appeared in the same variables that had already been detected when calculating the Weibull hazard function (for “grafting technique” and “height of rootstock”) (Table 5). Interpretation of the algebraic signs of the estimators is similar in both tables and is based on the interpretation of Table 5, which refers to the risk of death of the grafts (Cox, 1972). Therefore, the negative values of the estimators shown in Table 7 also indicate that the risk of death of the grafts decreases when the value of these variables increases, i.e., when xi > 1; i.e., when using the top cleft grafting technique and when the height of the rootstock is greater than the mean value observed (58.5 cm), the risk of death decreases and therefore, the probability of survival increases. On the other hand, the positive values of the variables “phenological stage of buds” and “diameter of scion” indicate that the risk of death increases when xi > 1 (grafts with buds at the beginning of sprouting and scions with diameters greater than the observed mean value, i.e., 11.4 mm).

The effects of the other seven grafting variables considered in the present study (Table 1) were also evaluated in the aforementioned study (Pérez-Luna et al., 2020b), and it was found that these variables did not significantly affect the risk of graft death. The significant effects observed in the present study contrast with these previous findings, which can be attributed to the fact that on this occasion the climatic conditions inside the greenhouse were controlled, thus reducing the variation caused by factors not evaluated. The survival of the grafts can therefore be more directly attributed to their response to the treatments and to the grafting variables evaluated. In a grafting experiment with Araucaria angustifolia, Wendling, Stuepp & Zuffellato-Ribas (2016) obtained survival greater than 20% after 130 days of evaluation for rootstocks between 80 and 100 cm in height; our findings indicate that the survival of P. engelmannii grafts is improved by using rootstocks taller than 55.8 cm.

Using scions with diameter larger than the rootstock diameter can cause localized graft incompatibility. This effect was visualized by Sweet & Thulin (1973), who reported this type of incompatibility in grafts of Pinus radiata D. Don, as the cambium did not coincide between the grafted organs. On the other hand, Pérez-Luna et al. (2020b) observed that when using short rootstocks of P. engelmannii (less than 30 cm), the graft was also short (less than 15 cm height in average), hampering establishment of a protective microenvironment and subsequent management of the grafts.

Pérez-Luna et al. (2019) also used the Cox’s proportional hazards model and the associated hazard ratio to evaluate the effect of the phenological stage of the scion buds (end of latency and beginning of sprouting), observing that this factor did not have a significant influence on the risk of graft mortality. The difference in the hazard ratio due to the effect of the phenological stage of the scion buds in the present study relative to the aforementioned research can mainly be attributed to the differences in management of the greenhouse conditions, as in the previous study the climate control was not used.

Conclusions

A total of two survival models used in medical studies proved useful tools for evaluating the success of P. engelmannii grafts. Thus, the Weibull’s accelerated failure time model and the Cox’s proportional hazards model and their respective hazard ratios were validated for use in predicting the survival rate (risk of death) as a function of the factors considered, such as the grafting technique, the phenological stage of scion buds and some other grafting variables inherent to the scions and rootstocks. Although the Cox’s proportional hazards model provided a better fit to the data, the use of the Weibull’s accelerated failure time model is also recommended, as it enabled reliable prediction of the estimated graft survival time after the end of the evaluation period. The best grafting technique for asexual propagation of P. engelmannii proved to be the top cleft method, and the best phenological condition for the scion buds was at the end of the latency period. Grafting was more successful with scions of diameter smaller than 11.4 mm. In addition, in order to reduce the risk of death of top cleft and side veneer grafts of P. engelmannii, the use of rootstocks taller than 58.5 cm is recommended. The results of the present study can serve as a guide for decision-making for grafting in the studied species.

Supplemental Information

Supplemental Information 1 Raw data.

Click here for additional data file.

The authors are grateful to Andrea Losoya Simental for help with English language editing. We also thank Rosa Elvira Madrid Aispuro, Silvia Salcido Ruiz and Manuel González Romero for assistance with grafting activities.

Additional Information and Declarations

Competing Interests

Author Contributions

Data Availability

Christian Wehenkel is an Academic Editor for PeerJ.

Alberto Pérez-Luna conceived and designed the experiments, performed the experiments, analyzed the data, prepared figures and/or tables, authored or reviewed drafts of the paper, and approved the final draft.

José Ciro Hernández-Díaz conceived and designed the experiments, performed the experiments, analyzed the data, authored or reviewed drafts of the paper, and approved the final draft.

Christian Wehenkel conceived and designed the experiments, performed the experiments, analyzed the data, authored or reviewed drafts of the paper, and approved the final draft.

Sergio Leonel Simental-Rodríguez conceived and designed the experiments, authored or reviewed drafts of the paper, and approved the final draft.

Javier Hernández-Velasco conceived and designed the experiments, authored or reviewed drafts of the paper, and approved the final draft.

José Ángel Prieto-Ruíz conceived and designed the experiments, performed the experiments, analyzed the data, authored or reviewed drafts of the paper, and approved the final draft.

The following information was supplied regarding data availability:

The raw data are available in the Supplemental Files.

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
