# Peer review of "Graft survival of Pinus engelmannii Carr. in relation to two grafting techniques with dormant and sprouting buds"

_PeerJ, doi:10.7717/peerj.12182_

## Round 0.1 · original submission · Major Revisions

Authors are requested to provide major revisions.

·

Basic reporting

The main issue is:

Most of the manuscript sections, as noted in the text, need to be significantly improved in English writing, with better order and organization for the sentences used. The reviewer made all necessary corrections and improvements and highlighted them in the various parts of the manuscript. All suggested corrections must be implemented, and the highlighted errors must be corrected.

Experimental design

No comment.

Validity of the findings

The main issue is:

The discussion section should be greatly improved. In other words, a more in-depth discussion is required for the majority of the results presented. Please take note of the highlighted issues and respond to the questions in the comments.

Additional comments

Dear respected author,

Thank you for your excellent scientific research. It is a valuable study, but it requires revisions based on the comments provided to you in the manuscript and this review form. All the issues raised should be addressed.

Sincerely,

·

Basic reporting

no comment. I have listed the review results in "General comments for the author".

Experimental design

no comment. I have listed the review results in "General comments for the author".

Validity of the findings

no comment. I have listed the review results in "General comments for the author".

Additional comments

In my opinion, this article is a very excellent study that used statistical methods to validate the effectiveness of grafting as one of the methods selected to propagate Pinus engelmannii. Therefore, I hope that this article will be published and will be seen by many readers. I have read this article many times, but I do not have many comments on it. If I have any comments, they are the following two points. These are minor comments.
1. In Tables 3 and 6, the authors examine the model selection of several factors. It is quite understandable that GT and PSB have a significant impact, but is HR or DS an essential factor in estimating graft survival? If the authors have a clear idea, please put it in the "Discussion" section.
2. The main purpose of this paper is to examine the factors that should be considered for grafting. However, statistical method selected is one of the important parts that reinforce the originality of this article. Therefore, I feel that the "Discussion" section should contain more information on the selection of statistical models. For example, the part corresponding to lines 319-330 in the "Results" section could be included in the "Discussion" section. Could the authors please reconsider this?

---

## Round 0.2 · Minor Revisions

Unfortunately the original Editor on this paper has resigned from the Board and is no longer available to make a decision. In addition, a reviewer who had promised their review is now very late, and so we do not believe we will receive their review.

Therefore we are returning the comments from Reviewer 1 for you to make the minor changes they have requested. We assume that once you resubmit, we will be able to assign an Editor from the Board to make the final Accept decision.

·

Basic reporting

The authors addressed all of the questions and made all of the necessary changes.

Experimental design

The experimental design is appropriate.

Validity of the findings

In the perfect form.

Additional comments

Dear respected author,

Thank you for your efforts in revising the manuscript in an appropriate manner. Annotated pdf file contains only minor writing errors in the results and discussion sections. Please fix these mistakes.

Sincerely,

---

## Round 0.3 · Minor Revisions

I am now the Editor of your paper, since the prior Editor is unavailable, as the journal explained to you in the previous communication.

I thank you for all the corrections and changes. I believe that the new manuscript improved greatly after the modifications, and in this version the overall presentation and the English is very good.

I have only 6 minor, mainly editorial, details before the paper can be accepted.

1) In line 285, something is missing. It says: “only the phenological stage of the buds was significant after the Bonferroni correction”, but from the table, that variable is the only one that is not significant. Perhaps you wanted to say “only the phenological stage of the buds was not significant after the Bonferroni correction”.
2) In line 286: An explanation of what are the patterns of the values of the Weibull’s risk function shown in Table 5 is needed, along with an interpretation of the HRW values, explaining in which cases is lower and in which higher, as these values are not necessarily obvious to most readers.
3) In Table 5, the “Weibull’s risk function” is called “Weibull hazard function”; I suggest changing “risk” in line 286 to “hazard” (as originally defined in Methods, line 208 and following). Also, change it to “hazard” in line 412 and elsewhere in the text if needed.
4) In Figure 2, in the figure legend explain what are the error bars in the figure, i.e., standard deviation, standard error?
5) In Figure 2B, you need to change “Phenolical” to “Phenological”.
6) In Figure 3, also in the figure legend, explain what the error bars in the figure are.

---

## Round 0.4 · accepted · Accept

Thank you for the changes and corrections in the manuscript and in the figures.

I consider that the manuscript is now ready for publication, and want to acknowledge your efforts, and congratulate all the authors for this interesting contribution.